

# Spatial variation in allometric growth of invasive lionfish has management implications

Juan Carlos Villaseñor-Derbez[*] and Sean Fitzgerald[*]

Bren School of Environmental Science and Management, University of California, Santa Barbara, Santa Barbara, CA, United States of America
[*] These authors contributed equally to this work.

## ABSTRACT

Lionfish (*Pterois volitans/miles*) are an invasive species in the Western Atlantic and the Caribbean. Improving management of invasive lionfish populations requires accurate total biomass estimates, which depend on accurate estimates of allometric growth; sedentary species like lionfish often exhibit high levels of spatial variation in life history characteristics. We reviewed 17 published length-weight relationships for lionfish taken throughout their invasive range and found regional differences that led to significant misestimates when calculating weight from length observations. The spatial pattern we observed is consistent with findings from other studies focused on genetics or length-at-age. Here, the use of *ex situ* parameter values resulted in total biomass estimates between 76.2% and 140% of true observed biomass, and up to a threefold under- or overestimation of total weight for an individual organism. These findings can have implications for management in terms of predicting effects on local ecosystems, evaluating the effectiveness of removal programs, or estimating biomass available for harvest.

## INTRODUCTION

Lionfish (*Pterois volitans/miles* complex) are an invasive species in the Western Atlantic Ocean and Caribbean Sea, likely introduced through release of aquarium-kept organisms (*Betancur-R et al., 2011*). Lionfish are the first invasive marine vertebrates established along these coasts (*Schofield, 2009*; *Schofield, 2010*; *Sabido-Itza et al., 2016*), and they have established populations in coral reefs, estuaries, mangroves, hard-bottomed areas, and mesophotic reefs (*Barbour et al., 2010*; *Jud et al., 2011*; *Muñoz, Currin & Whitfield, 2011*; *Claydon, Calosso & Traiger, 2012*; *Andradi-Brown et al., 2017*; *Gress et al., 2017*). Their presence has been labeled as a "major marine invasion" because they threaten local biodiversity, spread rapidly, and are difficult to manage (*Hixon et al., 2016*).

A substantial amount of research describes lionfish impacts throughout their invaded range. A meta-analysis by *Peake et al. (2018)* showed that invasive lionfish prey on at least 167 different species across the tropical and temperate Western Atlantic. Their feeding

Corresponding author
Juan Carlos Villaseñor-Derbez,
jvillasenor@bren.ucsb.edu

behavior and high consumption rates can reduce recruitment and population sizes of native reef-fish species, and can further endanger reef fish (*Green et al., 2012*; *Rocha et al., 2015*; but see *Hackerott et al. (2017)* for a counterexample). For example, field experiments showed that lionfish in the Bahamas led to reduced recruitment of native fishes by nearly 80% over a five-week period (*Albins & Hixon, 2008*), and prey fish biomass declined by 65% over two years as lionfish biomass increased along Bahamian coral reefs (*Green et al., 2012*). However, trophic impacts of lionfish can be minimized if their local biomass is controlled by culling (*Arias-Gonzalez et al., 2011*).

Governments and non-profit organizations have sought to reduce lionfish densities through removal programs and by incentivizing its consumption (*Chin, Aiken & Buddo, 2016*). In some cases, these have shown to significantly reduce—but not quite eliminate—lionfish abundances at local scales (*de Leon et al., 2013*; *Sandel et al., 2015*). Complete eradication of lionfish through fishing is unlikely because of their rapid recovery rates and ongoing recruitment to shallow-water areas from persistent populations in mesophotic ecosystems (*Barbour et al., 2011*; *Andradi-Brown et al., 2017*). However, promoting lionfish consumption might create a level of demand capable of incentivizing a stable fishery while controlling shallow-water populations, thus creating alternative livelihoods and avoiding further negative effects to local biota.

The feasibility of establishing fisheries through lionfish removal programs has been extensively evaluated through field observations and empirical modeling (*Barbour et al., 2011*; *Morris, Shertzer & Rice, 2011*; *de Leon et al., 2013*; *Johnston & Purkis, 2015*; *Sandel et al., 2015*; *Usseglio et al., 2017*). Determining the feasibility of such initiatives requires modeling the change in biomass in response to changes in fishing mortality (i.e., culling). A common way to model this is via length-structured population models, where fish lengths are converted to weight to calculate total biomass (*Barbour et al., 2011*; *Côté et al., 2014*; *Andradi-Brown et al., 2017*). The allometric length-weight relationship is thus an essential component of these models, but this relationship can vary across regions as a response to biotic and abiotic conditions (*Johnson & Swenarton, 2016*).

Outcomes of previous studies suggest lionfish are likely to exhibit spatial heterogeneity in the length-weight relationship for both behavioral and biological reasons. Important life history characteristics such as growth or natural mortality rates are often spatially variable for fish that exhibit sedentary behavior (*Gunderson et al., 2008*; *Hutchinson, 2008*; *Wilson et al., 2012*; *Guan et al., 2013*), and in fact, high levels of site fidelity and small home ranges are two primary reasons why culling programs are effective in reducing local adult lionfish populations (*Fishelson, 1997*; *Kochzius & Blohm, 2005*; *Jud & Layman, 2012*; *Côté et al., 2014*). Genetic analysis of lionfish also identified two genetically distinct invasive subpopulations between the Western Atlantic and the Caribbean, suggesting the existence of spatially explicit biological differences between populations as well (*Betancur-R et al., 2011*). Site-specific studies that calculate the length-weight relationship of lionfish report variable estimates, and these differences may be increasingly important when estimating the potential effectiveness of lionfish culling programs (*Barbour et al., 2011*; *Morris, Shertzer & Rice, 2011*; *Côté et al., 2014*; *Johnston & Purkis, 2015*). However, the influence of using *ex situ* parameters when estimating the length-weight relationship remains unexplored.

Our objective was to quantify the magnitude of error caused by using *ex situ* parameter values when estimating lionfish weight from length observations. In this study, we calculated and reported the first length-weight relationship for lionfish in the central Mexican Caribbean using previously collected *in situ* observations ($n = 109$; *Villaseñor-Derbez & Herrera-Pérez (2014)*). We then estimated lionfish weight in this area using previously published length-weight relationships for lionfish populations from ten locations across the Western Atlantic, Gulf of Mexico, and Caribbean. By comparing these weight estimates to our *in situ* length-weight observations, we showed that using *ex situ* parameter values resulted in up to a threefold under- or overestimation of lionfish weight and estimated total biomass ranged between 76% and 140% of observed total biomass.

## METHODS

We reviewed 12 published studies and obtained 17 length-weight relationships for the Western Atlantic ($n = 2$), Gulf of Mexico ($n = 7$), and Caribbean ($n = 8$), Table 1, Fig. 1. Study sites included North Carolina, the Northern and Southern Gulf of Mexico, the Southern Mexican Caribbean, the Bahamas, Little Cayman, Jamaica, Bonaire, Puerto Rico, and Costa Rica (*Barbour et al., 2011*; *Darling et al., 2011*; *de Leon et al., 2013*; *Fogg et al., 2013*; *Dahl & Patterson, 2014*; *Edwards, Frazer & Jacoby, 2014*; *Toledo-Hernández, 2014*; *Sandel et al., 2015*; *Aguilar-Perera & Quijano-Puerto, 2016*; *Sabido-Itza et al., 2016*; *Sabido-Itzá, Aguilar-Perera & Medina-Quej, 2016*; *Chin, Aiken & Buddo, 2016*). We have access only to the summarized information published in these studies—not the raw data authors used to make length-weight calculations. We collected information on sex differentiation, location, length and depth ranges, and sampling methods from each study when available. Only two studies reported sex-specific length-weight parameters (*Aguilar-Perera & Quijano-Puerto, 2016*; *Fogg et al., 2013*), so we assumed data were reported for both sexes combined in all other studies. Reviewed studies presented information for organisms ranging from 25–475 mm in Total Length (*TL*) and were obtained at depths between 0.5 m and 57 m. Four studies explicitly stated that their organisms were sampled with pole spears (*Dahl & Patterson, 2014*; *Aguilar-Perera & Quijano-Puerto, 2016*; *Chin, Aiken & Buddo, 2016*; *Sabido-Itzá, Aguilar-Perera & Medina-Quej, 2016*), and six studies mentioned that some of their organisms were obtained with pole spears (or other type of harpoon) but also hand-held nets or fish traps (*Barbour et al., 2011*; *Fogg et al., 2013*; *Edwards, Frazer & Jacoby, 2014*; *Toledo-Hernández, 2014*; *Sandel et al., 2015*; *Sabido-Itza et al., 2016*). Two studies did not specify how organisms were sampled (*Darling et al., 2011*; *de Leon et al., 2013*).

We also used data from 109 lionfish sampled by *Villaseñor-Derbez & Herrera-Pérez (2014)*, who used hand nets and numbered bottles to collect Total Length (*TL*; mm) and Total Weight (TW; g) for organisms from 10 sampling sites along the central Mexican Caribbean coast in the summer of 2010 (Table S1). Sampling locations included wall and carpet reefs at depths between 5.7 m and 38.1 m. The use of hand nets prevented any weight loss due to bleeding and allowed better representation of small sizes by avoiding gear selectivity. Organisms were euthanized via pithing.

**Table 1 Summary of 18 allometric growth parameters available for lionfish in the invaded range from peer-reviewed literature and this study.**
All parameters have been adjusted to convert from millimeters to grams.

| Region | Sex | $n$ | $a$ | $b$ | $R^2$ | Reference |
|---|---|---|---|---|---|---|
| Western Atlantic | B | 774 | 2.90 | 2.89 | – | *Barbour et al. (2011)* |
| Western Atlantic | B | – | 0.25 | 3.29 | – | *Darling et al. (2011)* |
| GoM | B | 934 | 0.21 | 3.34 | 0.98 | *Dahl & Patterson (2014)* |
| GoM | B | 472 | 0.29 | 3.30 | 0.95 | *Aguilar-Perera & Quijano-Puerto (2016)* |
| GoM | F | 67 | 0.12 | 3.47 | 0.95 | *Aguilar-Perera & Quijano-Puerto (2016)* |
| GoM | M | 59 | 0.42 | 3.23 | 0.95 | *Aguilar-Perera & Quijano-Puerto (2016)* |
| GoM | B | 582 | 0.14 | 3.43 | 0.99 | *Fogg et al. (2013)* |
| GoM | M | 119 | 0.27 | 3.31 | 0.97 | *Fogg et al. (2013)* |
| GoM | F | 115 | 0.68 | 3.14 | 0.94 | *Fogg et al. (2013)* |
| Caribbean | B | 458 | 3.60 | 2.81 | – | *Sandel et al. (2015)* |
| Caribbean | B | 419 | 2.80 | 2.85 | 0.87 | *Chin, Aiken & Buddo (2016)* |
| Caribbean | B | 1,450 | 2.30 | 2.89 | 0.92 | *de Leon et al. (2013)* |
| Caribbean | B | 1,887 | 0.30 | 3.24 | 0.97 | *Edwards, Frazer & Jacoby (2014)* |
| Caribbean | B | 2,143 | 0.52 | 3.18 | 0.99 | *Sabido-Itza et al. (2016)* |
| Caribbean | B | 227 | 0.80 | 3.11 | 0.96 | *Toledo-Hernández (2014)* |
| Caribbean | B | 449 | 0.23 | 3.25 | 0.97 | *Sabido-Itzá, Aguilar-Perera & Medina-Quej (2016)* |
| Caribbean | B | 368 | 0.32 | 3.19 | 0.98 | *Sabido-Itzá, Aguilar-Perera & Medina-Quej (2016)* |
| Caribbean | B | 109 | 0.32 | 3.23 | 0.98 | This study |

**Notes.**

n, Sample size, Sex specifies whether data was presented for Females (F), Males (M), or both sexes combined (B); a, scaling parameter (presented in $\times 10^{-5}$); b, exponent.

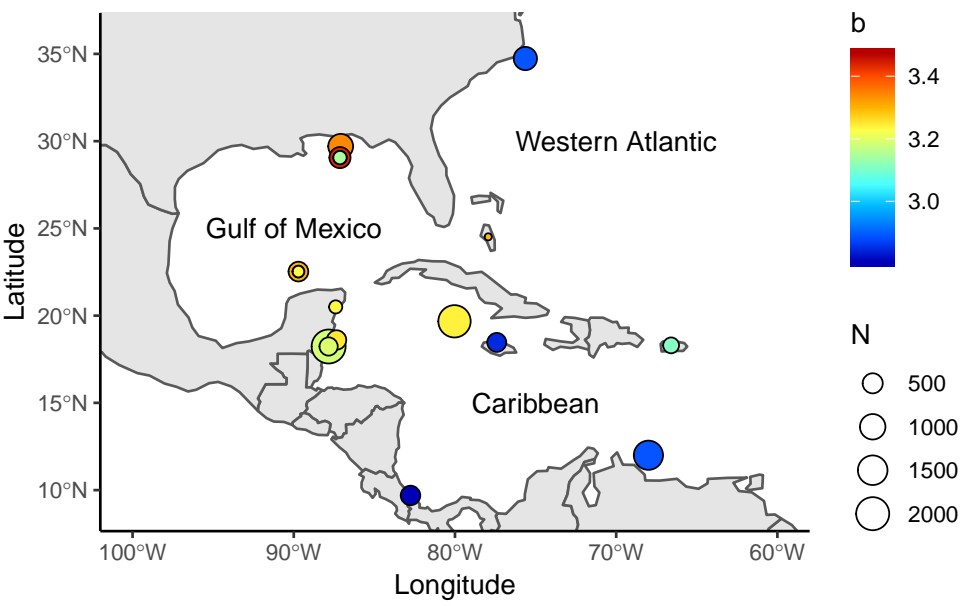

**Figure 1 Locations where allometric growth parameters of lionfish (*Pterois spp*) have been reported.**
Circle sizes indicate sample size from each study, colors indicate the *b* coefficient from (1).

The weight-at-length relationship for lionfish in the central Mexican Caribbean was calculated with the allometric growth function:

$$TW = aTL^b \tag{1}$$

where $a$ is the ponderal index and $b$ is the scaling exponent or allometric parameter. We linearized the equation using $log_{10}$-transformation and estimated the coefficients using an Ordinary Least Squares Regression with a heteroskedastic-robust standard error correction (*Zeileis, 2004*). Coefficients were tested with a two-tailed Student's $t$-test, and the significance of the regression was corroborated with an $F$-test. Some of the reviewed studies (Table 1, Fig. 1) inconsistently defined $a$ as either the ponderal index from Eq. (1) or the $y$-intercept from the linearized log-transformed equation. Other studies incorrectly reported parameters as mm-to-g conversions when they were in fact cm-to-g conversions. We standardized each study by converting coefficients and report all parameters as $TL$ (mm) to TW (g) conversions.

We obtained a total of 18 parameter pairs by combining length-weight parameters extracted from the literature and the additional pair calculated here (Table 1). Recall that the objective of this study is not to describe differences in the length-weight relationship between populations (which would require access to raw data), but rather to assess how *ex situ* parameter values influence the accuracy of weight estimates for lionfish, using the central Mexican Caribbean as a case study. Using each of the 18 parameter pairs, we estimated $TW$ from the $TL$ observations collected in the central Mexican Caribbean ($n = 109$, with $TL$ ranging from 34 mm to 310 mm) and divided predicted weights by known observed weights to obtain a simple measure of over- or underestimation. Difference in mean weight ratios were tested with an analysis of covariance (ANCOVA):

$$R_{i,j} = \tilde{\mu} + \alpha_j + \beta TL_{ij} + e_{ij} \tag{2}$$

where $R_{ij}$ is the weight ratio for the $i$-th organism obtained with parameters from the $j$-th study, $\tilde{\mu}$ is a constant for all individuals, $a_j$ is the treatment effect (i.e., the difference induced by each study), $TL_{ij}$ is the covariate (i.e., Total Length for the $i$-th subject in the $j$-th group) with slope $\beta$, and $e_{ij}$ is the error term of the regression. Ratios were logit-transformed prior to analysis, and a *post-hoc* Tukey's test was used to identify groups where mean ratios did not differ. All analyses were performed in R version 3.5.2 (*R Core Team, 2018*). Raw data and code used in this work are available on github at github.com/jcvdav/lionfish_biometry.

## RESULTS

The length-weight relationship for organisms from the central Mexican Caribbean (Fig. 2) resulted in coefficient values of ($a = 3.205 \times 10^{-6}$) and $b = 3.235$ ($R^2 = 0.977$, $F_{1,107} = 6928.67$; $p < 0.001$). The allometric factor (b) was significantly different from $b = 3$ ($t_{107} = 6.04$; $p < 0.001$), corroborating that lionfish present allometric growth. The length-weight coefficients estimated here were within the range identified by studies from other regions (Fig. 3, Table 1).

ANCOVA results revealed significant differences in our predicted weight ratios for the central Mexican Caribbean when using each of the different pairs of parameters

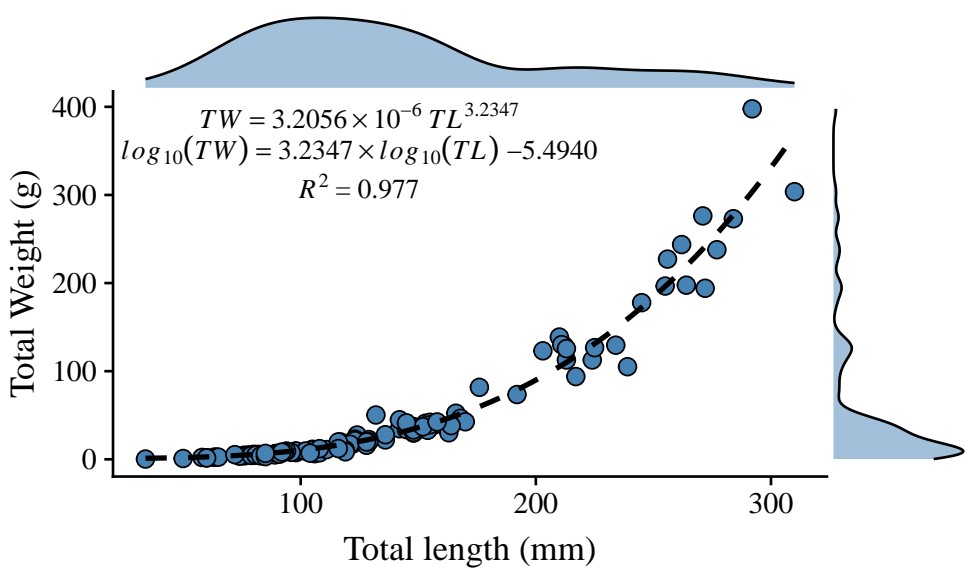

**Figure 2** **Length-weight relationship for 109 lionfish sampled in the central Mexican Caribbean.** Points indicate samples, dashed black line indicates curve of best fit, marginal plots represent the density distribution of each variable.

($F_{17,1943} = 24.96$; $p < 0.001$; Fig. 4). For example, the actual observed weights of the 109 lionfish from the Central Mexican Caribbean had a mean ± SD of 52.56 ± 76.58 g. However, if we used allometric parameter values from Banco Chinchorro in the Caribbean to predict weights from our observed length observations, we estimated a mean ± SD of 40.37 ± 58.74 g (*Sabido-Itzá, Aguilar-Perera & Medina-Quej, 2016*). If we similarly used parameter values from North Carolina in the Western Atlantic to estimate lionfish weights in the Central Mexican Caribbean, we found a mean ± SD of 73.76 ± 96.11 g (*Barbour et al., 2011*). Weights predicted from these extreme parameters correspond to mean predicted-to-observed weight ratios of 0.80 ± 0.19 and 1.76 ± 0.50 (mean ± SD), respectively. Furthermore, largest errors for individual organisms collected in the central Mexican Caribbean resulted in ratios of 0.36 and 3.51 (i.e., the tails of each violin in Fig. 4). If we examined biomass (i.e., summing across all 109 organisms) instead of mean ratios, total biomass estimates were 76.2% (4,363.53 g) and 140% (8,039.96 g) of true observed biomass (5,729.34 g). Parameters for this study estimate total biomass at 98% of observed biomass (Fig. 5). These misestimates come from the two most extreme sets of parameters, but results varied consistently across locations (Figs. 4 and 5). Overall, the use of *ex situ* parameters led to significantly erroneous estimates of individual weight and total biomass for lionfish.

Tukey's post-hoc test showed that weight ratios for the central Mexican Caribbean differed from those obtained with parameters from the Western Atlantic in North Carolina (*Barbour et al., 2011*), and most sites in the Caribbean and the Gulf of Mexico (Tukey's HSD $p > 0.05$). The only sites where weight ratios did not differ from the central Mexican Caribbean were Little Cayman (*Edwards, Frazer & Jacoby, 2014*), Bahamas (*Darling et al.,*
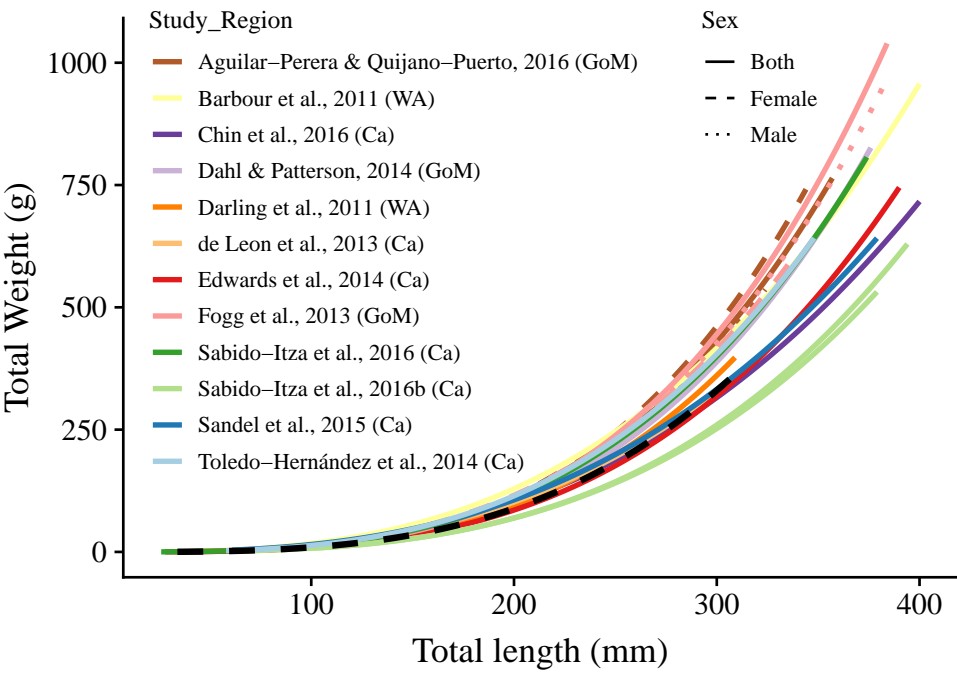

**Figure 3** **Length-weight relationships ($n = 18$) for 12 studies and this study.** The curves are shown for the range of lengths reported in each study (See Table S2); when ranges were not present, we use the ones found in this study (34 mm–310 mm). Colors indicate studies from which the parameters were extracted. Dotted, dashed, and solid lines show models for males, females, and combined sexes, respectively. Letters in parentheses indicate if the study comes from the Gulf of Mexico (GoM), Western Atlantic (WA), or Caribbean (Ca). The dashed black line represents the relationship estimated in this study. There are two solid green lines for *Sabido-Itza et al. (2016)*, one for each of the two sites for which they report parameters. A log–log version of this figure is presented in Fig. S1.

*2011*), and the Northern Gulf of Mexico (*Dahl & Patterson (2014)*; Tukey's HSD $p > 0.05$). All weight estimates using parameters from the Gulf of Mexico and Western Atlantic were higher than observed values, and only parameters from the Caribbean produced weights smaller than observed (Fig. 4). The regional average ± SD of predicted-to-observed weight ratios from these three regions were $1.24 \pm 0.309$, $1.41 \pm 0.523$, and $1.20 \pm 0.423$ for the Gulf of Mexico, Western Atlantic, and Caribbean, respectively. This suggests that the smallest errors are observed when using parameters from other locations in the Caribbean.

## DISCUSSION

Our results suggest that lionfish exhibit highly variable, spatially heterogeneous allometric relationships across their invaded range, and that this variation may be relevant for managing invasions. Moreover, we show that the use of *ex situ* parameter values may lead to highly biased weight and total biomass estimates. Our comparison of observed weights to those predicted with locally-informed parameters and *ex situ* parameters showed that weight of an individual lionfish can be overestimated by more than threefold, highlighting the need to use local information. Here we discuss the implications of our findings, possible shortcommings in our analyses, and highlight potential future research directions.

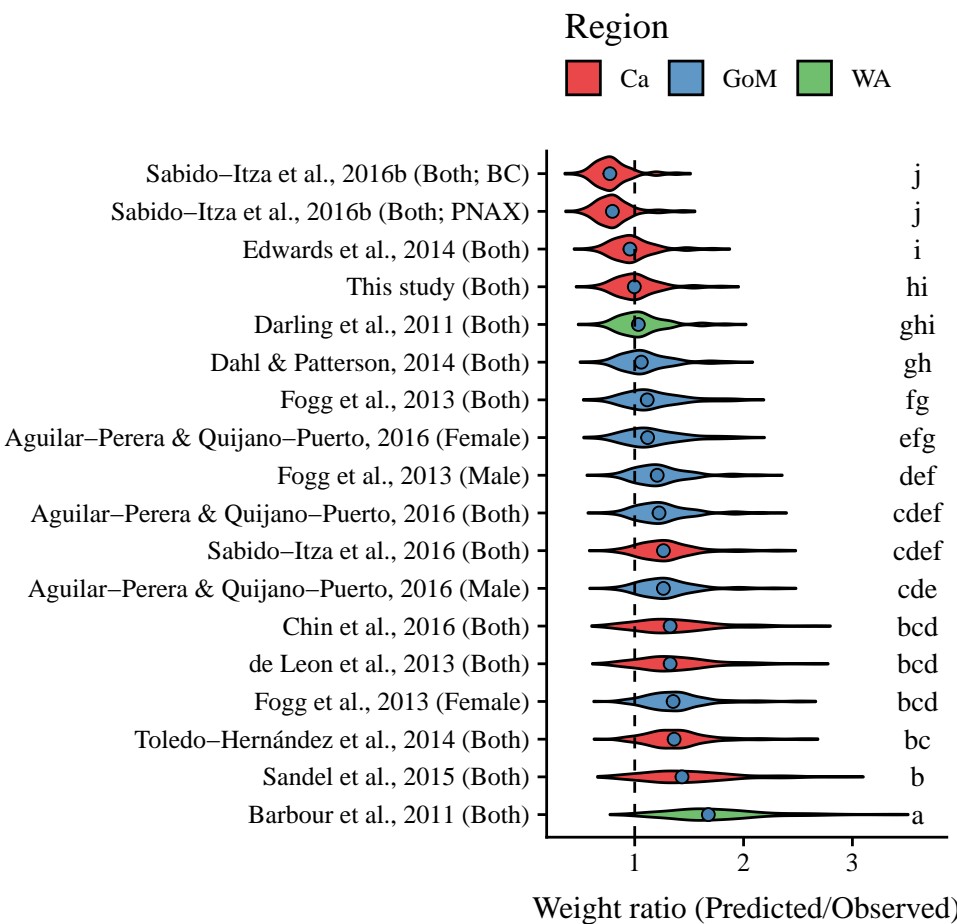

**Figure 4  Violin plot of predicted-to-observed weight ratios when applying each of 18 different pairs of allometric parameters to the 109 lionfish collected in the central Mexican Caribbean.** Sex is indicated in parentheses. Blue circles indicate median values and like letters indicate values that do not differ significantly. For Sabido-Itza et al., 2016b, BC and PNAX make reference to Banco Chinchorro and Parque Nacional Arrecifes de Xcalak, two sites for which they report parameters.

Differences in length-weight relationships have traditionally been highlighted as potential pitfalls to fishery management. For example, *Wilson et al. (2012)* showed that small-scale variations in length-at-age and fishing mortality in other Scorpaeniformes translate to differential landings, effort, and catch per unit effort in the live fish fishery of California, and that these differences must be taken into account in management plans. The lionfish case poses the opposite scenario, where the manager desires to eradicate the species. To accurately gauge both the effectiveness of lionfish removal efforts and the resources needed to successfully manage an invasion, we must acknowledge and understand regional biological differences in important variables such as allometric growth parameters.

We detected substantial differences in weight-at-length between organisms from the Caribbean, Gulf of Mexico, and Western Atlantic. Groupings of predicted-to-observed weight ratios identified in our *post hoc* testing aligned with the spatial distribution of

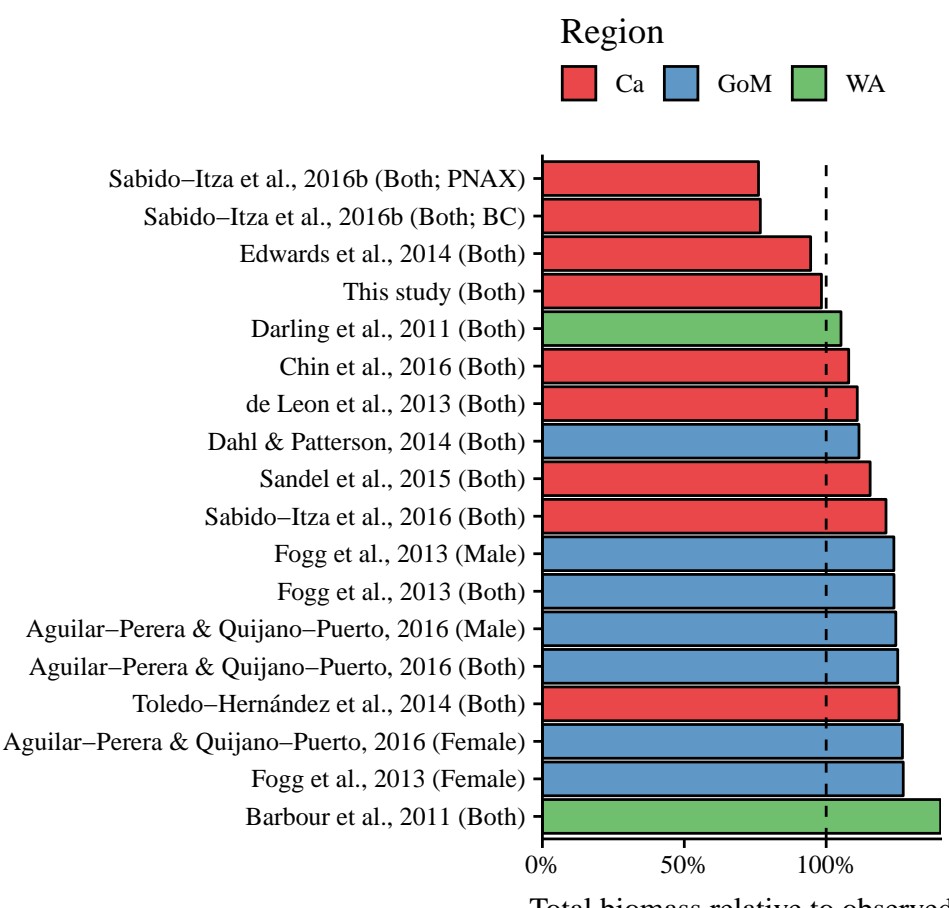

**Figure 5** **Estimated total biomas relative to observed biomass (5,729.34 g) for 18 pairs of allometric parameters.** Sex is indicated in parentheses. For *Sabido-Itza et al. (2016)*, BC and PNAX make reference to Banco Chinchorro and Parque Nacional Arrecifes de Xcalak, two sites for which they report parameters.

the examined studies, suggesting that these differences may be mediated by space. These regional allometric differences mirror similar patterns in length-at-age of lionfish across both their invaded and native regions (*Pusack et al., 2016*). Variation may be driven by genetics or by organisms' exposure to distinct environmental conditions. For example, *Betancur-R et al. (2011)* used mitochondrial DNA to demonstrate the existence of two distinct population groups, identified as the "Caribbean group" and "Northern Group", and *Fogg et al. (2015)* alternatively suggested that length-at-age differences may be driven by the environment.

One might be inclined to attribute all variation in the lionfish length-weight relationship to the spatial origin of these parameters. However, samples from the 12 studies included here were not only collected in different locations, but also at different points in time, time of the year, and across different depth and size ranges (See Table S2 for an extended version of Table 1). The magnitude of the bias discovered in this study and our lack of understanding the sources driving spatial variation for lionfish highlights the need to simultaneously

collect length-weight information across the invaded range to test for spatially-induced patterns and to link these findings to previously suggested environmental and genetic structures. Such an endeavor would provide insight into lionfish biology and better inform management. However, while we could not evaluate how these factors influenced length-weight estimates from previous studies without raw data, we still show that a lack of locally-calculated parameters can induce significant bias when calculating weight from length observations. We demonstrate the importance of using *in situ* parameters to obtain accurate weight estimates regardless of the underlying mechanisms driving variation between populations.

Applying parameter estimates to lengths outside the range of lengths originally used to estimate the parameters may also induce error. Our smallest observed organism was 34 mm in *TL*, and only two studies estimated parametrs with smaller organisms (*Edwards, Frazer & Jacoby, 2014*; *Sabido-Itza et al., 2016*). By contrast, our largest organism had a *TL* of 310 mm, which is well within the range of all other studies (the next smallest maximum length was 325 mm; see Table S2). Due to the power function describing the allometric relationship (i.e., Eq. (1)), the error in weight estimates is larger when extrapolation is done for lengths that are larger than the maximum length used to estimate the parameters. Our estimates are therefore conservative because we only used parameter pairs from other studies to estimate weights for lionfish up to 310 mm in the Central Mexican Caribbean, well within the range of lengths for which other parameters were estimated.

## CONCLUSION

The results presented here have key implications for management. For example, *Edwards, Frazer & Jacoby (2014)* simulated a lionfish culling program under two scenarios, one using length-at-age and length-to-weight parameters from North Carolina and one using parameters from Little Cayman. Their results show that using different parameters caused up to a four-year difference in the time required for the simulated lionfish population to recover to 90% of its initial biomass after removals ceased.

Here, we show that using one set of length-weight parameters versus another for a given length can result in more than a threefold under- or overestimation of total weight for individual fish, and that total biomass estimates may range between 76% and 140% of true observed biomass. These differences become especially important when allocating resources for lionfish removal programs, incentivizing lionfish fisheries as a source of alternative livelihoods, or estimating ecosystem impacts. Research efforts focused on invasive lionfish populations need to use parameters calculated for their region to the extent possible, or use different sets of parameters that provide appropriate upper and lower bounds in their results.

## ACKNOWLEDGEMENTS

We thank Nils Van Der Haar and Michael Doodey from Dive Aventuras as well as Guillermo Lotz-Cador who provided help to collect samples. We are grateful for comments raised

by the editor and two anonymous reviewers, which significantly increased the quality of this work.

### Funding

Juan Carlos Villaseñor-Derbez was supported by the Latin American Fisheries Fellowship Program and UCMexus-CONACyT Doctoral fellowship (CVU: 669403). The funders had no role in study design, data collection and analysis, decision to publish, or preparation of the manuscript.

### Grant Disclosures

The following grant information was disclosed by the authors:
Latin American Fisheries Fellowship Program and UCMexus-CONACyT Doctoral fellowship: (CVU: 669403).

### Competing Interests

The authors declare there are no competing interests.

### Author Contributions

- Juan Carlos Villaseñor-Derbez and Sean Fitzgerald analyzed the data, contributed reagents/materials/analysis tools, prepared figures and/or tables, authored or reviewed drafts of the paper, approved the final draft.

### Data Availability

GitHub: https://github.com/jcvdav/lionfish_biometry.

### Supplemental Information

Supplemental information for this article can be found online at http://dx.doi.org/10.7717/peerj.6667#supplemental-information.

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
