# Peer review of "Spatial variation in allometric growth of invasive lionfish has management implications"

_PeerJ, doi:10.7717/peerj.6667_

## Round 0.1 · original submission · Major Revisions

Both reviewers noted a need to consider different or additional analyses and to improve the writing of this contribution. Along with your revised ms., please include the text of both reviews in you point-by-point responses to the reviewers' comments.

Reviewer 1 ·

Basic reporting

The word choice needs to be consistent throughout the paper when referring to various items (see below). The word "major" is a subjective term and not appropriate to use in this manuscript. There is not a test for major or non"majorness", rather let the reader decide what is major. There are a few other word choice comments or clarity questions that I have, see below.

L 19: Using the word “major” is a subjective word and elicits an emotional response usually in those that read it. Please choose a less loaded word. Major to one person is not necessarily major to all people. Instead use substantial, significant, etc. Please make the appropriate changes throughout the manuscript as well ( results L142; discussion: L 175).
Introduction:
Line (L) 23: Capitalized “western” as was written in the Abstract and insert Ocean after “Atlantic”. As it reads now it sounds like the Atlantic Sea.
L24: replace “liberation” with “release”.
L 37: Albins and Hixon (2008) did their research in the Bahamas not Florida. Please correct.
L47: Replace impacts with “negative effects”.
L61: replace “know” with “known”.
Fig 1. To stay consistent throughout the manuscript please refer to the general region of the Atlantic ocean the same way. Throughout the manuscript it is referred to as the Wester Atlantic, North Atlantic, North-Western Atlantic. Choose one way to refer to this throughout all text and figures (see lines: 11, 23, 23, 81, 145 158). Not sure I found them all.

Methods:
L 82: The Bahamas are actually part of the Western Atlantic Ocean, not the Caribbean, please update your sample sizes.

L 115: You changed how you refer to grams, here you use (gr) where as you used (g) before (see lines 100, 113). For all references to grams, both in the text and figures, please use (g).

L 121: It is not clear to me what the TL ϵ (34,310) means. Can you re-write or clarify what is meant by this?

L125: Will you add the github URL so the readers can find the data and code. It was not clear where the raw data would be shared either.

Results:
L 134-135: Did you use all of the “different pairs of parameters” from all of the studies individually? Do I understand the right? If so can you adjust the words to be “each of the different pairs of parameters”?

L 134-135: It is not clear what you mean by the lowest and highest weight estimates. For what size did you estimate these? These are pretty low weights so It must have been a small length, why just do it for a small length? Would you assume the effect to be the same among at all lengths? I suspect not with the exponential relationship.

Fig. 3. Why are there two solid light green lines for the Sabido-Itza et al. 2016b? IN the legend for sex make sure that there are two dashed lines for female not just one short dashed line.

Fig. 4. In the caption please indicate what “BC” and “PNAX” mean for the Sabido-Itza 2016b refer to. It is not clear to the reader.

Discussion:
L 152: The phrase “variation is related to space” does not really say much. What do you mean by related to space? Do you mean that the is spatial heterogeneity in the parameters?

L 152: I am not sure what word you mean to use when you said: “we shot that”. Please choose a different word.

Experimental design

L 82-92: The authors do a good job indicating many of the differences among the studies, I wonder about the time of year, which might have a large effect on lionfish growth. One would expect both temperature and food availability to change. What time of year were these collected?

L 92-94: Why include the Fogg et al. (2013) data when you already know that it is likely to underestimate the weight due to the use of spineless weight. What effect did this have on the analysis using data that you know to be different than the other data? Interestingly, Fig. 3 shows that the Fogg data has some of the largest weights-at-length, not the smallest.

Validity of the findings

L 107-108: There appear to be quite a few differences in the sampling and processing methods among all of the studies (lines 82-92). Which the authors rightly discuss and support the conclusion that using local in situ, not ex situ parameters are critical to use. However, I wonder how this affects the data analysis. To use an ordinary least squares regression when all of the locations and study methods are likely to have an effect does not seem appropriate. I suggest using a mixed modeling technique with "study" as a random effect and the other parameters from equation (3) as the fixed effects, so running a LMM. If the error is not normal, as I suspect because you used a standard error correction you could then use a GLMM with a Gamma error distribution since the response variables are continuous and strictly positive. Doing this type of analysis would allow you to talk more specifically about the amount of variation the occurs based on the ex situ parameters to a site.

L175: Remove the word major from the main conclusion. Use either key, central, or fundamental. Major seems to be too strong and subjective of a word.

Additional comments

Overall this study is a nice review of the existing data on allometric growth of lionfish. I like using new data as a case study to show local and regional differences and highlights the need to collect data at the local level to understand how this invasion manifest across the invaded region. I am interested to see how using a different statistical test might be able to shed some light on what the actual variation is among sights and using a mixed-model method should help you estimate this.

Reviewer 2 ·

Basic reporting

See attached letter

Experimental design

See attached letter

Validity of the findings

See attached letter

Additional comments

See attached letter

Annotated reviews are not available for download in order to protect the identity of reviewers who chose to remain anonymous.

---

## Round 0.2 · accepted · Accept

Two previous reviewers were pleased with the revision and now find the manuscript largely acceptable for publication. Reviewer 1 has some minor issues that I would ask the authors to consider in drafting the final version of their ms while in production.

# Reviewer 1 ·

Basic reporting

Line (L) 26: The word “invasive” is redundant with the previous sentence. There are only invasive populations at these locations as indicated in the previous sentence.

L 36: delete the word “establishment”

L 140: The standard deviation reported is quite large, and would have a lower end stretch into the negative numbers. I assume this was because of the many small lionfish and few of the larger ones that were collected. Is this why the heteroskedastic-robust standard error correction from Zeileis (2004) (L 109) was used? If so can you indicate this around line 109?

L 153-156: Because the Bahamas are in the Western Atlantic (as indicated in Fig. 3), it is inaccurate to say, “ that weight ratios for the central Mexican Caribbean different from those obtained from the Western Atlantic” (L153-154), but “ratios did not differ” from the Bahamas (L156). Please adjust the wording.

L 165: Because this study did not evaluate directly different management practices I would adjust the wording to, “and that this variation may be relevant for managing invasions.”

L 182: I suggest switching the word “space” to “location”, which I think is more accurate for what the authors mean.

L190: I suggest changing “points in time” to “times of the year” to more specifically suggest seasonality in growth rates

Experimental design

Figure 3/Figure 5. There appears to be a potential separation in the allometric models between the northern sites GoM/WA and CA, albeit there is one CA site that appears to cluster with the GoM/WA, from one of the locations by Sabido-Itza (2016). (1) Did you look at all into clustering at a regional level instead of local and see how that would affect your predictions? Are the northern sites different than the Southern sites? Would this regional level be useful to use for management? (2) was there anything unusual about the sites for the Sabido-Itza (2016) sites that cluster closer to the GoM/WA, such as depth? I only bring this up because of what Bentacur-R et al. (2011) found with a Northern and Caribbean group difference from mitochondria DNA. If there is a difference that this study could report, it would be a documented phenotypic difference matching a mitochondria difference, which would be an important finding to report. I think it is worth looking into this to strengthen the results and potential impact of this manuscript.

To do this in a simple way, you could look at the order shown in Fig 5 and see how likely this order is by random chance, to cluster based on regional level. If you randomize the order of the local sites 1,000 or 10,000 times, then look at the number of times that you had this clustering at the regional level or more extreme.

Validity of the findings

no comment

Additional comments

Overall, the Authors greatly improved the manuscript and addressed all the comments very well. The take-home message that heterogeneity of lionfish growth within the invaded range is important to document, especially for management purposes and local adaptation. The authors do a good job of presenting their data and conclusions. Below are a few minor comments that should be addressed before the manuscript is published, but the manuscript is very close and is well written and presented. I know it might be a bit more work to address the comment about the data present in Fig 3/ Fig 5, but as a fellow lionfish research and interested in how this invasion has progressed, it should be looked at and hopefully without too much work.

Reviewer 2 ·

Basic reporting

The authors have clarified the language in their submission to meet the requests of the reviewers.

Experimental design

The authors have made the necessary changes to the analysis and presentation of their findings based on the reviewer suggestions.

Validity of the findings

The findings are valid and the data are robust. The contribution to the field is supported by their data.

Additional comments

Thank you for your thorough responses to each of the reviewer recommendations. The paper is much improved.